# *Agrobacterium tumefaciens*-Mediated Transformation of the Aquatic Fungus *Phialemonium inflatum* FBCC-F1546

**DOI:** 10.3390/jof9121158

**Published:** 2023-12-01

**Authors:** Jonghan Yoon, Youngjun Kim, Seoyeon Kim, Haejun Jeong, Jiyoon Park, Min-Hye Jeong, Sangkyu Park, Miju Jo, Sunmin An, Jiwon Park, Seol-Hwa Jang, Jaeduk Goh, Sook-Young Park

**Affiliations:** 1Department of Plant Medicine, Sunchon National University, Suncheon 57922, Republic of Korea; yoonjonghan99@gmail.com (J.Y.); nate9702@gmail.com (Y.K.); cbc6976@gmail.com (S.K.); wjdgowns98@naver.com (H.J.); jiyoonpak@gmail.com (J.P.); minhye1962@gmail.com (M.-H.J.); miju7188@gmail.com (M.J.); sminan201@gmail.com (S.A.); jiwongogo17@gmail.com (J.P.); swsw104@gmail.com (S.-H.J.); 2Interdisciplinary Program in IT-Bio Convergence System (BK21 Plus), Sunchon National University, Suncheon 57922, Republic of Korea; 3Fungi Research Team, Microbial Research Department, Nakdonggang National Institute of Biological Resources, Donam 2-gil 137, Sangju 37242, Republic of Korea; skpark79@nnibr.re.kr

**Keywords:** *Agrobacterium tumefaciens*-mediated transformation, fungal transformation, gene manipulation, *Phialemonium inflatum*

## Abstract

*Phialemonium inflatum* is a useful fungus known for its ability to mineralise lignin during primary metabolism and decompose polycyclic aromatic hydrocarbons (PAHs). However, no functional genetic analysis techniques have been developed yet for this fungus, specifically in terms of transformation. In this study, we applied an *Agrobacterium tumefaciens*-mediated transformation (ATMT) system to *P. inflatum* for a functional gene analysis. We generated 3689 transformants using the binary vector pSK1044, which carried either the hygromycin B phosphotransferase (*hph*) gene or the enhanced green fluorescent protein (*eGFP*) gene to label the transformants. A Southern blot analysis showed that the probability of a single copy of T-DNA insertion was approximately 50% when the co-cultivation of fungal spores and *Agrobacterium tumefaciens* cells was performed at 24–36 h, whereas at 48 h, it was approximately 35.5%. Therefore, when performing gene knockout using the ATMT system, the co-cultivation time was reduced to ≤36 h. The resulting transformants were mitotically stable, and a PCR analysis confirmed the genes’ integration into the transformant genome. Additionally, *hph* and *eGFP* gene expressions were confirmed via PCR amplification and fluorescence microscopy. This optimised transformation system will enable functional gene analyses to study genes of interest in *P. inflatum*.

## 1. Introduction

*Phialemonium* (Sordariales: Cephalothecaceae) is a cosmopolitan fungus belonging to the phylum Ascomycetes. It is found in various natural environments, including air, soil, sewage, plants, and water [1]. Several species of the *Phialemonium* genus cause opportunistic human infections; however, unlike these species, *P. inflatum* promotes plant growth [2] and its leaf endophytes protect seedlings against root-knot nematode infection [3,4]. Furthermore, *P. inflatum* reportedly mineralises lignin during primary metabolism [5], indicating its ability to degrade lignin-like recalcitrant polycyclic aromatic hydrocarbons (PAHs). Despite the potential usefulness of *P. inflatum*, a transformation system to explore its gene functions for biological applications is lacking.

Various transformation techniques have been developed for the gene manipulation and functional analysis of fungi including protoplast-mediated transformation and electroporation [6,7,8,9,10]. The *Agrobacterium tumefaciens*-mediated transformation (ATMT) system can be applied to budding yeast [11,12,13] as well as filamentous fungi [9,14,15,16,17,18,19,20,21,22,23]. It has the advantage of producing transformed organisms simply by mixing *Agrobacterium* cells with asexual spores, mycelia, and even fruiting bodies, without physically removing the cell wall or using cell-wall-degrading enzymes. Furthermore, because the Ti plasmid is randomly integrated into fungal genomes, it provides an advantage in forward genetics [9,15,23,24].

To apply the ATMT transformation system to fungi, several conditions must be considered [24,25,26,27,28]. Firstly, provided the ATMT technique is applicable to the target fungus, the next step is to determine the optimal transformation conditions. The first condition to consider is the application of acetosyringone (AS), which may not be necessary for certain fungi [29,30]. Additionally, the number of fungal spores and *Agrobacterium* cells during co-cultivation can affect the number of resulting transformed cells [24,25,26,27,28]. Another factor to consider is the duration of co-cultivation, because a supraoptimal duration can result in additional T-DNA insertions [27,28]. Therefore, it is crucial to determine the optimal conditions to establish an ATMT in a specific fungus by evaluating these factors.

In this study, we aimed to establish an efficient *P. inflatum* genetic manipulation system using an ATMT system and report an optimised transformation protocol for *P. inflatum* using the strain BCC-F1546, which was isolated from an aquatic fungus.

## 2. Materials and Methods

### 2.1. Fungal Strains, Growth Media, Culture Conditions, and Hygromycin B Sensitivity

The wild-type *P. inflatum* strain BCC-F1546 used in this study was obtained from the Nakdonggang National Institute of Biological Resources, Sangju, Korea. The wild-type stain and its transformants were cultured at 25 °C under dark conditions on potato dextrose agar (PDA, Difco Laboratories, Detroit, MI, USA) or in potato dextrose broth (PDB, Difco Laboratories). The wild-type strain, along with the transformants generated from it, were stored as spore suspensions in 15% glycerol at −80 °C. 

The bacterial strain *A. tumefaciens* AGL-1, carrying the binary vector pSK1044, was used in this study [28]. The bacterial strain was cultured at 28 °C for two days in Luria Bertani (LB) broth (Difco Laboratories) and stored in 15% glycerol at −80 °C.

To determine the optimal hygromycin B concentration for selecting transformants, wild-type *P. inflatum* strains were evaluated by dropping 5 μL spore suspension (1 × 10^3^/mL) on PDA amended with different hygromycin B concentrations (0, 20, 40, and 50 µg/mL). After incubating the PDA plates at 25 °C for 10 days, colony growth was evaluated. This test was performed in triplicate.

### 2.2. ATMT of P. inflatum

The protocol for the ATMT was based on previous studies [27,28,31] with slight modifications to suit *P. inflatum*. In brief, *A. tumefaciens* cells were cultured at 28 °C for two days with shaking (150 rpm) in minimal medium (MM) [31] containing kanamycin (50 μg/mL). To induce bacterial cells for fungal transformation, the cultured *A. tumefaciens* cells were transferred to induction medium (IM) [15] containing 200 μM AS following collection of the 5 mL cell culture using centrifugation (5000 rpm for 5 min). The transferred bacterial cells in IM medium were cultured at 28 °C in a shaking incubator for 6 h. 

For fungal transformation, conidia were obtained from the seven-day-old PDA culture. After adding 5 mL of sterilised distilled water (SDW) to the PDA culture, the PDA surface with *P. inflatum* spores was gently scraped with a spatula and an additional 5 mL of SDW was added. To remove the mycelia, the harvested spore suspension was filtered through two layers of sterilised Miracloth. The spore suspension was then centrifuged (5000 rpm for 5 min) to remove the mycelia, and 10 mL of SDW was added, vortexed, and centrifuged again three times to remove the mucilage on the spore surface. The prepared spores were counted using a haemocytometer. Before transformation, the prepared spores were diluted with 5 mL SDW.

For co-cultivation, 100 μL of *P. inflatum* spores (10^4^, 10^5^, 10^6^, and 10^7^) and 100 μL of induced *A. tumefaciens* cells (OD = 0.6) were mixed and evenly spread onto cellulose membrane (cellulose nitrate; 47 mm diameter and 0.45 μm pores; Whatman Ltd., Maidstone, UK) on co-cultivation medium amended with or without 200 µM AS.

After co-cultivation at 28 °C for 24 h, 36 h, and 48 h, only cellulose membranes were transferred from the co-cultivation medium to selection media containing hygromycin B (50 µg/mL) and cefotaxime (250 µg/mL). One week after transfer, the mycelia that grew on the cellulose membrane were transferred to a 24-well plate (SPL, Seoul, Republic of Korea) containing selection media with hygromycin B (50 µg/mL) and cefotaxime (250 µg/mL). For the transformants selected for the second time in this way, approximately 10 pieces of mycelia with a diameter of 5 × 5 mm were placed in a solution containing 15% glycerol and then cultured at 25 °C for three days. The cultures were then stored at −80 °C for long-term preservation.

### 2.3. Genomic DNA Extraction

Fungal genomic DNA was extracted from mycelia cultured in 5 mL PDB in a 6-well plate (SPL, Republic of Korea) at 25 °C for one week. The grown wild-type mycelia and transformants were harvested, freeze-dried, immersed in liquid nitrogen to freeze, and then pulverised into fine powder using a Mini-Beadbeater-96 (1001, Biospec products, Bartlesville, OK, USA). Genomic DNA was prepared using the NucleoSpin Plant kit (Macherey-Nagel, Düren, Germany) according to manufacturer’s instructions. The amount of extracted DNA was measured using a NanoDrop-1000 spectrophotometer (Thermo Fisher Scientific, Wilmington, DE, USA).

### 2.4. Southern Blot Analysis

Southern blot analysis was performed to determine the copy number of inserted T-DNA in the transformants. Genomic DNA (2 µg) was digested with *Hind* III and separated via gel electrophoresis using a 0.7% agarose gel at 40 V for 4 h in 1% Tris-acetate-EDTA buffer. The DNA fragments in the agarose gel were transferred to Hybond N^+^ membrane (Amersham International, Little Chalfont, UK) using 20× SSC (1.5 M NaCl and 0.15 M sodium citrate) and then the DNA fragment on Hybond N^+^ membrane was cross-linked via ultraviolet (UV) light.

For probe preparation for Southern blot analysis, the *hph* gene was obtained via PCR amplification using pSK1044 [28] as a template and the primers HygB_F (5′-TCAGCTTCGATGTAGGAGGG-3′) and HygB_R (5′-TTCTACACAGCCATCGGTCC-3′). The resulting fragments were labelled using a direct chemiluminescence labelling system via random priming (AlkPhos Direct^TM^ Labelling and Detection system, GE Healthcare, Chicago, IL, USA). 

Southern blot analyses and signal detection were performed using the AlkPhos Direct^TM^ Labelling and Detection system (GE Healthcare) according to the manufacturer’s instructions with slight modifications. Hybridisation was performed overnight at 55 °C. Hybridised blots were washed twice with primary and secondary washing buffers according to manufacturer’s instructions. Signals were detected using the CDP-Star^TM^ detection reagent, and the signal was exposed to hyperfilm^TM^ ECL (GE Healthcare) for 2 h.

### 2.5. PCR Amplification for Transformant Confirmation

PCR amplification of the *hph* and *eGFP* genes was performed using 20 ng genomic DNA and 10 pmol of each primer using the i-StarMAX II PCR master mix system (iNtRON Biotechnology Inc., Seongnam, Republic of Korea). For primers for *hph* gene amplification, HygB_F and HygB_R (described above) were used. For *eGFP* gene amplification, primers eGFP_F (5′-ATGGTGAGCAAGGGCG-3′) and eGFP_R (5′-TTACTTGTACAGCTCGTC-3′) were used. The amplification conditions were as follows: initial denaturation at 94 °C for 3 min; 30 cycles of 30 s of denaturation at 94 °C, 30 s of annealing at 56 °C, and 1 min of elongation at 72 °C; and finally, 5 min of elongation at 72 °C and maintenance at 4 °C. The PCR products were confirmed via electrophoresis on a 0.8% agarose gel, stained with Ecodye^TM^ DNA staining solution (Solgent Co. Daejeon, Republic of Korea) and visualised under UV light.

### 2.6. eGFP Expression Observation in Transformants

Purified transformants via single spore isolates were observed using a Zeiss Axio Imager A1 fluorescence microscope (Carl Zeiss, Oberkochen, Germany) with the following filter settings: excitation of 470/40 nm and emission of 525/50 nm.

### 2.7. Identification of T-DNA Insertion Sites via Inverse PCR

Inverse PCR (i-PCR) [32] was performed to recover genomic sequences flanking each inserted T-DNA. A total volume of 7 μL (100 ng of genomic DNA) was placed in a PCR tube, and 2.3 μL of 10 × *Taq* I enzyme buffer (Takara, Tokyo, Japan), 2.3 μL of bovine serum albumin, and 11.1 μL of SDW were added. Finally, 2 units of *Taq* I restriction enzyme (Takara, Tokyo, Japan) were added to the PCR tube for a final volume of 23 μL. The melting temperature was set to 65 °C, the PCR tube was placed in the thermocycler, and then it was processed for 3 h. 

To ligate the DNA, 5 μL was transferred to a new PCR tube. To each tube, 2 μL of 10 × T4 DNA ligase buffer (Takara, Tokyo, Japan), 12.7 μL of SDW, and 0.3 μL (corresponding to 2 units) of T4 DNA ligase (Takara, Tokyo, Japan) were added. The thermocycler was set to 4 °C, and tubes were placed in the PCR machine and processed overnight. 

After transferring 3 μL of self-ligated genomic DNA to new PCR tubes, 10 μL of i-Star MAXII mix was added to each tube. In addition, 1 μL each of 10 pmol/μL RB3 (5′-CCCTTCCCAACAGTTGCGCA-3′) and RBn1R (5′-TTTTCCCAGTCACGACGTTGTAA-3′) primers and 5.0 μL of sterilised distilled water were added to adjust the total volume to 20 μL to perform i-PCR.

PCR conditions were as follows: 94 °C for 3 min, followed by 32 cycles of 94 °C for 3 min, 94 °C for 30 s, 62 °C for 30 s, and 72 °C for 3 min, followed by a final PCR step at 72 °C for 5 min. The PCR products were confirmed on a 0.8% agarose gel. The PCR products obtained were sequenced and analysed for T-DNA insertion as previously described [24].

### 2.8. Mitotic Transformant Stability

The mitotic stability of hygromycin B resistance of three randomly selected transformants was tested by growing them on PDA without hygromycin B for three generations at 25 °C. Finally, the growing mycelial edge of the colony was transferred to PDA containing 50 μg/mL of hygromycin B.

### 2.9. Data Analysis

Normal one-way analysis of variance was used for statistical analyses in this study, and the significance of differences among treatments was determined using Duncan’s multiple range test. All statistical analyses were performed using SPSS version 22.0 (SPSS Inc., Chicago, IL, USA). Data shown are means ± standard deviation. Significance was defined at *p* < 0.05.

## 3. Results

### 3.1. ATMT Establishment for P. inflatum

The *hph* gene was used as a marker to select transformants. To use *hph* as a selection marker, we determined the dosage of hygromycin B for the wild-type *P. inflatum* FBCC-F1546 strain. When *P. inflatum* was inoculated on PDA medium supplemented with 0, 20, 40, and 50 μg/mL of hygromycin B, the wild-type fungal mycelia were completely inhibited with 50 μg/mL of hygromycin B (Figure 1A), indicating that the appropriate hygromycin B concentration was 50 μg/mL to screen for *P. inflatum* transformants.

ATMT may only be possible when AS is present in fungi [15,20,25,26]. We tested whether the presence or absence of AS affects the efficiency of ATMT in *P. inflatum*. As shown in Figure 1B, 200 µM AS was essential for the ATMT of *P. inflatum* during the induction stage of *A. tumefaciens* and the subsequent co-cultivation stage. 

To determine the appropriate number of spores for its efficient transformation, we used different numbers of spores. When 1 × 10^3^ spores per plate were used for the ATMT, 19.0 ± 1.0 transformants were generated. In addition, when 1 × 10^4^ and 10^5^ spores were added to each plate, 101.3 ± 23.8 and 103.7 ± 27.2 transformants were generated, respectively, which is approximately five times more than when 1 × 10^3^ spores were seeded per plate. The highest number of transformants was produced when 1 × 10^6^ spores were applied, which was twice as many as when 1 × 10^4^ or 10^5^ spores were used (Figure 1C). The results showed that to maximise the transformation efficiency of *P. inflatum*, the largest number of transformants could be obtained when 1 × 10^6^ spores were applied.

Next, to determine the effect of the co-culture time on the ATMT, we divided the co-culture time into 24, 36, and 48 h. The same number of spores (1 × 10^5^) was used per plate. As a result, 5.3 ± 2.1, 77.0 ± 12.0, and 149.0 ± 21.9 transformants were obtained when co-cultured for 24, 36, and 48 h, respectively (Figure 1D). These results indicated that 48 h was the optimum co-cultivation time for obtaining the largest number of transformants.

### 3.2. Analysis of T-DNA Insertion among the Generated Transformants

We showed that the number of transformants increased with increasing co-cultivation time (Figure 1D). However, this result does not indicate that T-DNA is preferentially integrated as a single copy into the *P. inflatum* genome. To determine the effect of the co-cultivation time, we randomly selected 24 transformants from each of the co-culture time points (24 h, 36 h, and 48 h) and confirmed the number of T-DNA insertions by a Southern blot analysis (Figure 2A). The single-copy T-DNA integration was confirmed to be 54.0%, 45.8%, and 35.5% after 24, 36, and 48 h under co-cultivation, respectively (Figure 2B). This result indicated that co-cultivation for 24 h was sufficient to obtain single-copy T-DNA integration in *P. inflatum*.

### 3.3. Confirmation of the Integration of the hph and eGFP Genes in the P. inflatum Genome

It is important to confirm the presence of both the *hph* and *eGFP* genes carried by T-DNA in the genome of *P. inflatum* using PCR amplification. In this study, we randomly selected 15 transformants to identify the genes in their genomes. In the wild-type strain, neither *hph* (Figure 3A, upper picture) nor *eGFP* (Figure 3B, lower picture) was amplified, whereas in the transformants, both genes (1 kb for the *hph* gene and 0.7 kb for the *eGFP* gene) were amplified, indicating that all 15 randomly selected transformants were T-DNA-inserted transformants.

Green fluorescence under a microscope was used to identify the transformants. All 15 transformants were examined for eGFP expression under a fluorescence microscope. Strong green fluorescence was observed in all the transformants, whereas no fluorescence was detected in the wild-type strain. Figure 3B shows the bright-field (DIC mode) and EGFP observations from four representative transformants (T-1, T-7, T-10, and T-14) and the wild-type strain. 

### 3.4. Identification of Genomic Sequences Flanking Inserted T-DNA in the Generated Transformants

I-PCR testing and subsequent sequencing were performed to identify the genomic location of the T-DNA insertion in the 15 selected transformants. After i-PCR testing, the agarose gel analysis showed PCR products of different sizes, ranging between 0.3 and 1 kb (Figure 3C). A sequence comparison among the 15 flanking regions suggested that the T-DNA was randomly inserted without preferential sequence contexts (Table 1). Truncated T-DNA was not detected in any of the 15 randomly selected transformants (Table 1).

### 3.5. Stability of Transformants in P. inflatum

We assessed the mitotic stability of the transformants after subculturing randomly selected transformants (T-1, T-7, and T14 in Figure 3B) for three rounds in PDA medium without hygromycin B. Finally, we transferred the hygromycin B-resistant transformants to a medium containing hygromycin B (50 µg/mL). All three transformants maintained resistance to hygromycin B, suggesting that the inserted T-DNA was mitotically stable.

## 4. Discussion

In order to harness valuable fungi for large-scale industrial processes, molecular genetic tools are essential. Indeed, various products are being commercially produced from *Aspergillus niger*, *A. oryzae*, and *A. sojae*. The main substances include glucoamylase, citric acid, cellulose, protease, and kojic acid [33]. These industrial products are used in food, beverages, pharmaceuticals, and cosmetics, among other applications. It is important to establish a genetic transformation system in order to obtain fungal strains with a higher efficiency for the production of these products. However, the transformation efficiency is generally low in filamentous fungi [34]. In particular, the application of the transformation technique via polyethylene glycol (PEG)-mediated protoplast transformation and restriction enzyme-mediated integration (REMI) with protoplasts are unsuccessful if the fungal cell wall is not sufficiently digested. ATMT has been successfully applied for the genetic transformation of a wide variety of fungal species, including industrial fungi (*Saccharomyces cerevisiae*, *Trichoderma reesei*, and *Aspergillus* spp.), mushrooms (*Pleurotus eryngii*, *Agaricus bisporus*, *Coprinopsis cinereus*, and *Volvariella volvacea*), and plant pathogenic fungi (*Magnaporthe oryzae*, *Fusarium verticillioides*, *F. oxysporum*, and *Verticillium dahliae*) [20,26,35].

The fungus *P. inflatum* has tremendous potential to produce very useful metabolites, with nematocidal, plant growth promotion, and PAH degradation activities [3,4,5]. In particular, the ability to degrade PAHs such as lignin and phenanthrene has been known to be present in white rot fungi (*Trametes versicolor*, *Lentinula edodes*, and *Phanerochaete chrysosporium*) [5]. Kluczek-Turpeinen et al. reported that *P. inflatum* can produce laccases without peroxidase activity [5]. The development of a transformation system for *P. inflatum* may help us to understand the production of various useful substances within this fungus. Because no transformation system was available for any *P. inflatum* strain, the primary aim of this study was to develop a *P. inflatum* transformation system.

We successfully developed and applied a transformation method to the conidia of *P. inflatum* via the ATMT method and generated 3689 hygromycin-resistant transformants that stably maintained the resistant cassette. In the previous study, uninucleate conidia in filamentous fungi were considered more suitable for transformation than multinucleate conidia [28]. We observed that the conidia of *P. inflatum* produce a relatively high number of transformants within a short co-cultivation time compared to that of other fungi [14,16,17,23,36,37,38,39,40]. Possibly, the insertion of T-DNA from the ATMT may have occurred more rapidly and easily in *P. inflatum*, which has single-nucleate conidia, compared to other fungi with multinucleated conidia. 

ATMT has not been previously used in this fungus as a tool for insertional mutagenesis. Thus, the T-DNA from the binary vector pSK1044 was efficiently transformed and correctly expressed in the genome. Previous studies on ATMT in ascomycetous fungi reported that T-DNA insertion favours single-copy integration [15,25,26]. This finding suggests that ATMT is a useful transformation strategy in forward genetics [9,15,24,25,41]. If the number of available transformants with a higher probability of single-copy integration increases, this will be the optimal condition for the ATMT of the target fungi. As shown in Figure 1D, the yield of transformants was highest after 48 h of co-cultivation and was >30 times higher than that after 24 h of co-cultivation. Such a short co-cultivation period results in a lower number of transformation events compared to those after 36, 48, or 72 h [13,15,28,29,37,42,43].

Regardless of the transformation efficiency rate, the phenotype of a gene deletion-mutation cannot be accurately tested when gene replacement is performed through homologous recombination via an ATMT, as extra copies of T-DNA may be inserted into the genome, potentially interfering with major traits. Thus, it was necessary to reduce the co-cultivation time to <24 h to increase single-copy T-DNA insertion (Figure 2B). As shown in Figure 2B, the probability of single-copy T-DNA insertion after 48 h decreased by 17% compared to that after 24 h. Our data imply that functional genomic studies on *P. inflatum* should be conducted under co-cultivation conditions <24 h to prevent the insertion of extra copies of T-DNA. Because the genome of *P. inflatum* FBCC-F1546 is currently being sequenced, we will use the optimal ATMT protocol developed here for the functional genetic analysis of *P. inflatum*.

We found that both exogenous genes, *hph* and *eGFP*, were effectively expressed in *P. inflatum*. We also found that transformants with *hph* and *eGFP* gene expression under the control of the *Aspergillus nidulans* trpC and *Cochliobolus heterostrophus* GAPD promoters, respectively, exhibited stable and constitutive expression in this fungus. This result demonstrates that both promoters can be used to express genes of interest in this fungus. Although two promoters have been conventionally used for gene overexpression, including genes that are generally expressed at low levels in a number of ascomycetous fungi [7,44,45], our data clearly indicated that these two promoters can be used for the overexpression of *P. inflatum*.

## 5. Conclusions

This study aimed to establish an efficient transformation system for *P. inflatum* to facilitate molecular genetic studies. We demonstrated that the ATMT method could be successfully applied to *P. inflatum*. To the best of our knowledge, this is the first report of genetic *P. inflatum* transformation via *Agrobacterium*. Unlike previous studies [13,15,28,29,37,43,44], the optimal conditions for single-copy T-DNA integration in *P. inflatum* were established. These conditions required a 24 h co-cultivation time with AS (200 µM) and 1 × 10^6^ spores per 5 cm plate. In addition, we successfully expressed the *hph* and *eGFP* genes in *P. inflatum* under the *A. nidulans* trpC and *C. heterostrophus* GAPD promoters, respectively. Ultimately, our findings will assist functional genetic analyses of *P. inflatum*.

## Figures and Tables

**Figure 1 jof-09-01158-f001:**
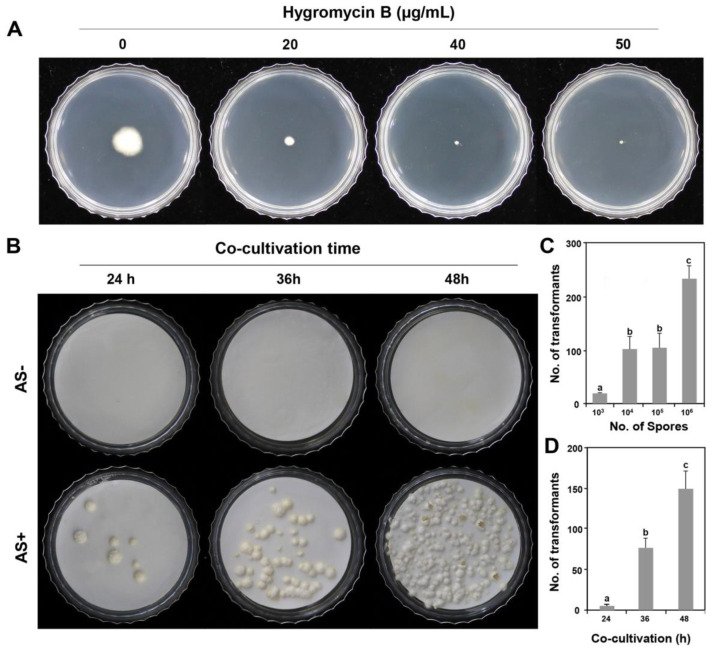
Hygromycin B sensitivity and factors affecting *Phialemonium inflatum*’s transformation efficiency. (**A**) *P. inflatum*’s sensitivity to hygromycin B after incubation for 10 days at 25 °C on potato dextrose agar (PDA) with 0, 20, 40, and 50 µg/mL of hygromycin B. (**B**) *Agrobacterium tumefaciens*-mediated transformation efficiency according to the presence or absence of AS and different co-cultivation periods. (**C**) Effect of the number of spores used in the transformation. (**D**) Effect of co-cultivation time; 10^6^ spores were used per plate. The data presented represent the mean of three independent experiments. Error bars indicate standard deviation. Analysis of variance (ANOVA) (**C**,**D**) was performed to test statistical differences at *p* ≤ 0.05. Tukey’s test was used to determine. The same letters in a column showed no significant difference.

**Figure 2 jof-09-01158-f002:**
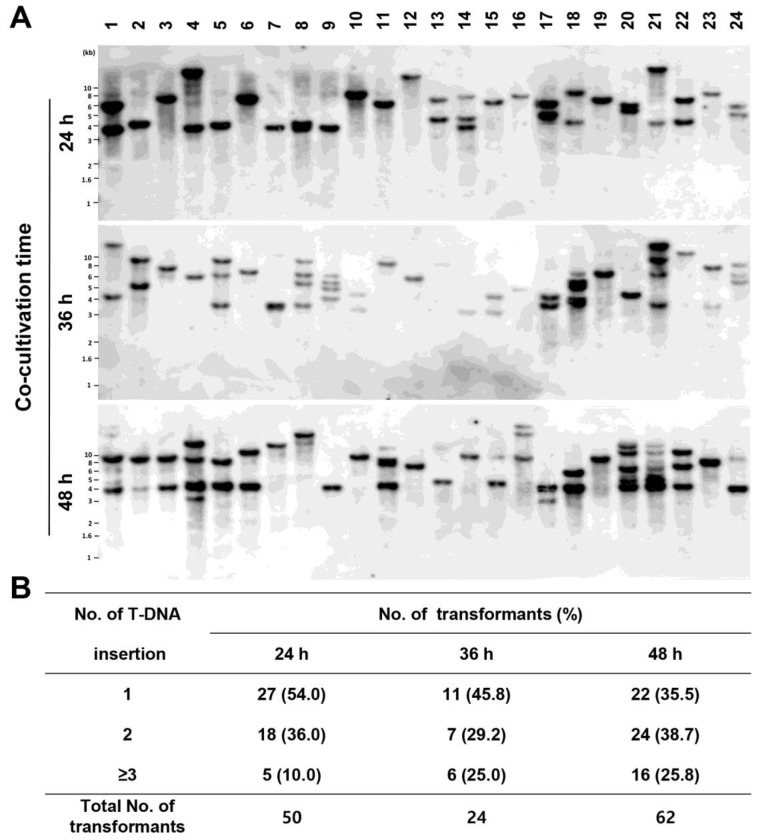
The transfer-DNA (T-DNA) copy number in *Phialemonium inflatum* transformants according to co-cultivation times of *P. inflatum* and *Agrobacterium tumefaciens*. (**A**) Southern blot analysis of the transformants. Genomic DNA of 24 randomly selected transformants from each co-cultivation period (24, 36, and 48 h) were probed with a labelled *hph* gene after digestion with *Hin*dIII, a restriction enzyme that does not cut the hph cassette. (**B**) T-DNA copy number distribution among 176 transformants according to different co-cultivation periods.

**Figure 3 jof-09-01158-f003:**
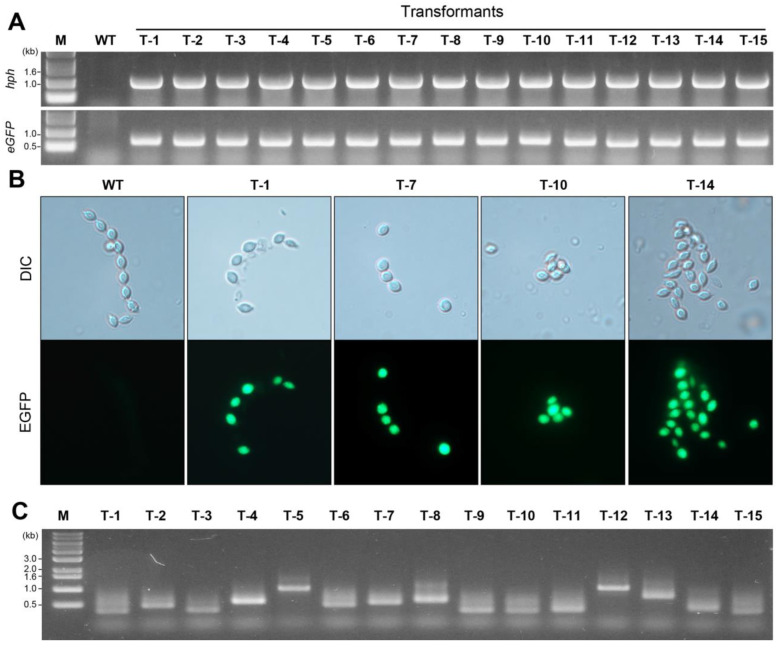
Confirmation of transformants and inverse PCR. (**A**) PCR amplification of the *hph* (upper) and *eGFP* (lower) genes in 15 randomly selected transformants. M indicates the DNA size marker. (**B**) EGFP expression in transformants. Bright-field (DIC mode) and fluorescence images of the wild-type strain and four transformants are shown. (**C**) Inverse PCR using 15 randomly selected transformants. M indicates the DNA ladder (kb).

**Table 1 jof-09-01158-t001:** Genomic sequences flanking the right border of inserted T-DNA in 15 *P. inflatum* transformants.

Transformants	T-DNA Copy Number	Sequences Flanking the Inserted T-DNA (5′-3′) ^a^
T-01	1	**GTTTAAACTATCAGTGTTTGA** ^b^ cggaacggatcatcaggctcgattt ^c^
T-02	1	**GTTTAAACTATCAGTGTTTGA**gtccttcatcctggactttcttcgatttta
T-03	1	**GTTTAAACTATCAGTGTTTGA**taaacaaattgacgcttaaaccacttaa
T-04	1	**GTTTAAACTATCAGTGTTTGA**aggttctgcgcccaccatgggcaggc
T-05	1	**GTTTAAACTATCAGTGTTTGA**caatttgctggtagtcctatcccgtccc
T-06	1	**GTTTAAACTATCAGTGTTTGA**cccagccaggcttttccccaggatacc
T-07	1	**GTTTAAACTATCAGTGTTTGA**tgggtggacctcgattttacgcacatat
T-08	1	**GTTTAAACTATCAGTGTTTGA**gagatcgattttacgcacatatgcgcat
T-09	1	**GTTTAAACTATCAGTGTTTGA**cgaggatgaggaagatgaaggcgac
T-10	1	**GTTTAAACTATCAGTGTTTGA**cccatcctctccagtaccaaactacctc
T-11	1	**GTTTAAACTATCAGTGTTTGA**ggtgtgtgtgtgtgagtgtgtctgggc
T-12	1	**GTTTAAACTATCAGTGTTTGA**cctacctgcctgtgtaccggagctcac
T-13	1	**GTTTAAACTATCAGTGTTTGA**cctcctgtctaattatccttgcttcgtttc
T-14	1	**GTTTAAACTATCAGTGTTTGA**tttttgttggtcgattttacgcacatatg
T-15	1	**GTTTAAACTATCAGTGTTTGA**cgttccgattcggaaaggaagaggg

^a^ Partial genomic sequences flanking the RB of inserted T-DNA in the analysed transformants are shown. ^b^ The bold capital letters correspond to the 21 bp right border sequences. ^c^ The small letters correspond to the last nucleotide of the host sequences.

## Data Availability

Data are contained within the article.

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
