# Peer review of "Agrobacterium tumefaciens-Mediated Transformation of the Aquatic Fungus Phialemonium inflatum FBCC-F1546"

_jof, 2023, doi:10.3390/jof9121158_

Round 1
Reviewer 1 Report
Comments and Suggestions for Authors
The ms by Yoon et al describes the Agrobacterium-mediated transformation of the fungus Phialemonium inflatum, which is of industrial importance. In the ms the experimental details are clearly described. I have only few critical remarks:
L202
I have never seen a report saying that ATMT is only possible when AS is absent. AS functions as a vir-inducer and is necessary for activating the regulatory VirA-VirG system of Agrobacterium, and is thus necessary for producing the T-strand in the bacterium. In some organisms transformation can occur without addition of AS, when such organism produces itself AS or another vir-inducer or grows on medium (for instance containing lignin) where such alternative vir-inducers can become available.
Line 263 and Table 1
In all cases the small part (TGA) of the RB repeat is present in the transformants, indeed showing the the T-DNA has not been truncted at this end. Formally the remaining part in bold of the sequence shown is not part of the RB as is said in note b. If “right border sequence” would be changed into “right end of the T-DNA” it would be correct.
Table1
What are the small letter sequences shown below the bold letter sequences? Are these sequences lost during transformation from the host genome, or are these the last nt of the host sequence that were determined. In the latter case maybe these sequences could be omitted.
Comments on the Quality of English LanguageEnglish largely ok.
Author Response
Dear reviewer,
We are grateful for the chance to improve our manuscript with resubmission. We have fully revised the manuscript. Our responses are highlighted in blue.
We hope this revised manuscript is acceptable for publication in Journal of Fungi. If you need any additional information or recommend any additional changes, please let me know. Once again, we appreciate your effort to improve the quality of our manuscript.
Sincerely,
Sook-Young Park
Reviewers' Comments to Author:
Reviewer 1
L202
I have never seen a report saying that ATMT is only possible when AS is absent. AS functions as a vir-inducer and is necessary for activating the regulatory VirA-VirG system of Agrobacterium, and is thus necessary for producing the T-strand in the bacterium. In some organisms transformation can occur without addition of AS, when such organism produces itself AS or another vir-inducer or grows on medium (for instance containing lignin) where such alternative vir-inducers can become available.
-- We removed this sentence. Instead, the following phrase was inserted in Line 202. “ ATMT may only be possible when AS is present in fungi.”
Line 263 and Table 1
In all cases the small part (TGA) of the RB repeat is present in the transformants, indeed showing the the T-DNA has not been truncted at this end. Formally the remaining part in bold of the sequence shown is not part of the RB as is said in note b. If “right border sequence” would be changed into “right end of the T-DNA” it would be correct.
-- Line 268: We changed “right border sequence” to “right end of the T-DNA”.
Table1
What are the small letter sequences shown below the bold letter sequences? Are these sequences lost during transformation from the host genome, or are these the last nt of the host sequence that were determined. In the latter case maybe these sequences could be omitted.
-- We revised it to “The small letters correspond to the last nucleotide of the host sequences with T-DNA insertion.”
Reviewer 2 Report
Comments and Suggestions for Authors
There are several points in the manuscript that must be corrected.
1. Line111: There is only the method of Genomic DNA extraction, not PCR analysis to confirm T-DAN insertion.
2. Fig.2B: the NO. of transformants in 24h is 27, 18, 5 in total 50, but the probability should be 54%, 36%, 10%, not 52.9%, 35.3%, 11.8%. So the number in Line227 shold be corrected.
3. eGFP was used as one gene name in Fig.3A, EGFP shold be used as protein name in Fig.3B, NOT GFP.
4. Are there references about Inverse PCR (i-PCR) in Line154?
Author Response
Dear reviewer,
We are grateful for the chance to improve our manuscript with resubmission. We have fully revised the manuscript. Our responses are highlighted in blue.
We hope this revised manuscript is acceptable for publication in Journal of Fungi. If you need any additional information or recommend any additional changes, please let me know. Once again, we appreciate your effort to improve the quality of our manuscript.
Sincerely,
Sook-Young Park
Reviewers' Comments to Author:
Reviewer 2
- Line111: There is only the method ofGenomic DNA extraction, not PCR analysis to confirm T-DAN insertion.
-- We changed “Genomic DNA extraction and PCR analysis to confirm T-DNA insertion” to “Genomic DNA extraction”.
- Fig.2B: the NO. of transformants in 24h is 27, 18, 5 in total 50, but the probability should be 54%, 36%, 10%, not 52.9%, 35.3%, 11.8%. So the number in Line227 should be corrected.
-- We have corrected our mistakes. For example, 52.9% was corrected to 54%. As mentioned in the manuscript, we described the number of single-copy T-DNA integrations after 24, 36, and 48 hours of co-cultivation, respectively. Thus, the percentages should be 54.0%, 45.8%, and 35.5%.
- eGFP was used as one gene name in Fig.3A, EGFP shold be used as protein name in Fig.3B, NOT GFP.
-- We changed “eGFP” to “EGFP” in line 249 and line 255.
- Are there references about Inverse PCR (i-PCR) in Line154?
-- We inserted a reference about Inverse PCR.